# Ultrasound-Assisted Extraction of Bioactive Compounds from Cocoa Shell and Their Encapsulation in Gum Arabic and Maltodextrin: A Technology to Produce Functional Food Ingredients

**DOI:** 10.3390/foods12020412

**Published:** 2023-01-15

**Authors:** Saeid Jafari, Zohreh Karami, Khursheed Ahmad Shiekh, Isaya Kijpatanasilp, Randy W. Worobo, Kitipong Assatarakul

**Affiliations:** 1Department of Food Technology, Faculty of Science, Chulalongkorn University, Bangkok 10330, Thailand; 2School of Agro-Industry, Mae Fah Luang University, Thasud, Chiang Rai 57100, Thailand; 3Department of Food Science, College of Agriculture and Life Sciences, Cornell University, Ithaca, NY 14853-5701, USA

**Keywords:** bioactive compounds, cocoa shell powder, functional foods, microencapsulation, ultrasonic extraction

## Abstract

In this study, the extraction of cocoa shell powder (CSP) was optimized, and the optimized extracts were spray-dried for encapsulation purposes. Temperature (45–65 °C), extraction time (30–60 min), and ethanol concentration (60–100%) were the extraction parameters. The response surface methodology analysis revealed that the model was significant (*p* ≤ 0.05) in interactions between all variables (total phenolic compound, total flavonoid content, and antioxidant activity as measured by 2,2-Diphenyl-1-picrylhydrazyl (DPPH) and ferric reducing antioxidant power (FRAP assays), with a lack of fit test for the model being insignificant (*p* > 0.05). Temperature (55 °C), time (45 min), and ethanol concentration (60%) were found to be the optimal extraction conditions. For spray-drying encapsulation, some quality metrics (e.g., water solubility, water activity) were insignificant (*p* > 0.05). The microcapsules were found to be spherical in shape using a scanning electron microscope. Thermogravimetric and differential thermogravimetric measurements of the microcapsules revealed nearly identical results. The gum arabic + maltodextrin microcapsule (GMM) showed potential antibacterial (zone of inhibition: 11.50 mm; lower minimum inhibitory concentration: 1.50 mg/mL) and antioxidant (DPPH: 1063 mM trolox/100g dry wt.) activities (*p* ≤ 0.05). In conclusion, the microcapsules in this study, particularly GMM, are promising antioxidant and antibacterial agents to be fortified as functional food ingredients for the production of nutraceutical foods with health-promoting properties.

## 1. Introduction

The use of bioactive compounds in pharmaceutical, food, and chemical industries, has necessitated the application of an eco-friendly technology for extracting active compounds from plants. Plant by-products, for example, contain a variety of bioactive chemicals with health-promoting properties [1]. Cocoa shell is considered as the by-product from cocoa processing which is typically underutilized and primarily used as fuel. On the account of health benefits, the cocoa shell has more potential than other dietary fiber sources [2]. According to Soares et al. [3], the presence of flavanols like epicatechins, catechins, and procyanidins, as well as alkaloids like caffeine and theobromine ensures good antioxidant potential in cocoa shell. Antioxidants are known to lower the risk of chronic heart disease and certain cancers [4,5].

Bioactive compounds can be extracted using a variety of conventional and novel techniques. Ultrasound-assisted extraction (UAE) has been widely used as a novel technique for the extraction of phytochemicals [6]. UAE increases the extraction rate and reduces the processing time when compared with other conventional methods. Several researchers have adopted response surface methodology (RSM) for the optimization of UAE to figure out the interactions and correlations of experimental factors [6,7]. A recent study employing RSM and ethanolic green extraction of phenolic compound from cocoa shells using variable extraction parameters yielded an optimized level of polyphenols [1]. The other study obtained the highest flavonoid level from Malaysian cocoa shell extracts by optimizing the UAE condition using RSM at three variables; the ethanol concentration (70–90% *v*/*v*), temperature (45–65 °C), and ultrasound irradiation time (30–60 min) [2].

Microencapsulation is a promising method for increasing the distribution of bioactive ingredients in foods using carriers, such as maltodextrin (MD) and gum arabic (GA) that can prevent digestion-related degradation, increase bioactivity/bioavailability for controlled release, and target administration in the consumers [8]. Microencapsulation of plant-based bioactive compounds has been accomplished through several techniques, including fluidized bed coating, inclusion complexation, complex coacervation, freeze-drying, spray-drying and extrusion [9]. However, amongst these popular encapsulation techniques, spray-drying is affordable, simple to apply, and produces high-quality powdered particles [7]. Choosing the right wall material/carrier is essential for microencapsulation. Only food-grade materials that have been certified as Generally Recognized as Safe (GRAS) or permitted by regulatory authorities like the Food and Drug Administration (FDA) and the European Food Safety Authority (EFSA) could be used for wall applications. Due to their high solubility, outstanding biocompatibility and safety, MD and GA are among the most frequently utilized bio-polymers in spray-drying encapsulation of bioactive chemicals [10]. In the past decade, numerous studies have been conducted on the characterizations of spray-dried microcapsules of phytochemical compounds from different cultivars [11,12,13]. In our recent research, we confirmed the encapsulation of bioactive compounds from mulberry leaf by spray-drying [6] and also used RSM to enhance their optimized extraction from makiang seed for orange juice fortification [14]. In addition, using the spray-drying process to microencapsulate bioactive compounds in fruit products could benefit the controlled release and safe target delivery of bioactives in the host. However, there is no information in the literature about the ultrasonic extraction, RSM optimization, and microencapsulation of cocoa shell polyphenols via the spray-drying process. Accordingly, the authors of the current work optimized an ethanolic green solvent extraction using RSM. Microencapsulated cocoa extracts were spray-dried and evaluated for antioxidant, antibacterial and microstructural properties for future application as functional food ingredients in the food industry.

## 2. Materials and Methods

### 2.1. Sample Preparation

A cocoa plantation in Nan, Thailand, provided the cocoa shell to the Department of Food Technology in Chulalongkorn University. The cocoa shells were washed, dried in a hot air oven at 60 °C for 48 h to achieve a moisture content of 5% after being cleaned under water to remove any dirt or dust, and powdered to pass through a 50-mesh sieve. CSP was then placed in an aluminum-laminated foil bag and stored at −20 °C for later use.

### 2.2. RSM Analysis and Optimizing UAE Extraction 

Response surface methodology (RSM) with a Box–Behnken Design (BBD) was used in this study (Table 1) to optimize three independent variables (X_1_, ethanol concentration; X_2_, temperature; X_3_, time) at three levels (−1, 0, 1). The responses were the total phenolic compound (TPC), total flavonoid content (TFC), 1,1-diphenyl-2-picrylhydrazyl (DPPH) and ferric-reducing antioxidant power (FRAP). The operational conditions for UAE in CSP were optimized using BBD with 17 experimental runs generated by RSM, which included ethanol concentration (X_1_: 60–100%), temperature (X_2_: 45–65 °C), and time (X_3_: 30–60 min) (Table 1 and Table 2).

For UAE extraction, 5 g of CSP samples was combined with 200 mL of ethanolic solvent. The UAE (Elmasonic bath P70H) was then conducted under various extraction conditions, as previously indicated. The extract was centrifuged at 6000 rpm for 15 min at room temperature (Centrifuge Kubota, series 6000, Osaka, Japan) and concentrated by a rotary evaporator at 40°C (Oilbath B-485, BÜCHI, Uster, Switzerland). The concentrated extract was then adjusted to 10 mL with distilled water and kept at 4 °C for further analysis. The procedure of UAE extraction was conducted as described previously [6].

### 2.3. Microencapsulation Experiment

The microencapsulation experiment was carried out as described previously [6]. In brief, MD, GA and a mixture of GA and MD (GMM) were mixed with CSP extract (40% *w/v* at a ratio of 1:2 *w*/*w*) in an agitator with constant stirring (SCILOGEX, model SCI550-S, Westmont, IL, USA) for 5 min and homogenized for 10 min in a high-speed blender (Ystral, model X10, Ballrechten-Dottingen, Germany). The solutions were then placed in a spray-dryer (Mobile Minor Niro-Atomizer, Søborg, Denmark) with a 155 °C inlet and 90 °C outlet temperature, feed temperature below 10 °C, and hot air flow rate 1.54 m^3^/min. The powders of GA microcapsule (GAM), MD microcapsule (MDM), and GMM were stored at −20 °C until analysis.

### 2.4. Determination of Bioactive Compounds (TPC and TFC) and Antioxidant Activity (DPPH and FRAP) in CSP Microcapsules

A total of 1 g of each microcapsule was dissolved in 10 mL of distilled water, vortexed for 3 min and kept in a hot shaking water bath (30 °C for 30 min), then centrifuged for 20 min at 4000 rpm. Then, the supernatant was collected for bioactive analyses. The procedure for functional screening was adopted as described previously [15,16,17]. In brief, TPC was assessed using the Folin–Ciocalleu method with gallic acid (0–0.5 mg/mL) as a standard. TFC was examined using the aluminum chloride colorimetry technique using a quercetin standard (0–1.6 mg/mL). A spectrophotometer was used to record the absorbance and difference between the DPPH solution and sample. The antioxidant activity was expressed as mM trolox equivalents/100 g dry weight. The FRAP values were determined using the difference in absorbance values at 593 nm between the FRAP solution and the CSP extract.

### 2.5. Physicochemical Properties of Microcapsules

A moisture analyzer (Mettler Toledo, Greifensee, Switzerland) was used to determine the moisture content (%). The water activity (a_w_) was measured using a water activity analyzer (model MS1, Novasina, Switzerland). The encapsulation efficiency (%) was measured as described by Saénz et al. [13]. The water solubility was conducted following the method of Sarabandi et al. [10]. In brief, 1 g of each powder sample was added to 100 mL of distilled water and homogenized with a magnetic stirrer (400 rpm for 4 min). The resulting solution was centrifuged for 4 min at 4000× *g*. A 25 mL portion of the supernatant was put into an aluminum cup that had been weighted before being dried in a 105 °C oven for 5 h. The water solubility was calculated using the weight of dried supernatant as a percentage of the initial powder.

At room temperature, a chroma meter Minolta CR-400 color meter that uses the CIE LAB system (L*, a*, and b*) was used for color determination. The morphology of microcapsules was examined using a scanning electron microscope at a magnification of 1000×. The particle size of the microcapsule was determined using a laser diffraction particle size analyser (Mastersizer 3000, Malvern Instruments Ltd., Malvern, UK).

### 2.6. Antimicrobial Activity Experiment

#### 2.6.1. Growth Condition 

Microbial cultures including *Escherichia coli* ATCC 25922, *Salmonella Typhimurium* ATCC 1331, *Staphylococcus aureus* ATCC 25923 and *Bacillus subtilis* ATCC 6633 were obtained from Thailand Institute of Scientific and Technological Research (TISTR; Pathum Thani, Thailand). Inoculum of each microorganism was prepared by inoculating with 10 mL sterile growth medium Muller Hinton Broth (MHB), Muller Hinton Agar (MHA). The cultures were grown on MHB and incubated at 37 °C overnight. 

#### 2.6.2. Antimicrobial Screening by Disc Diffusion Method 

Freshly prepared bacterial culture 100 µL was pipetted out in the center of a sterile Petri dish. MHA was poured into a Petri plate along with the inoculum and gently stirred. After solidification, wells were drilled into inoculated agar plates and fitted with a sterile disk. Then, on each disk, 20 µL of sample extract was added. The plates were left for 30 min to disseminate the sample extracts on the inoculated agar plates. The plates were then incubated for 18 h at 37 °C. After the incubation time, the zone of inhibition (including the well width) was measured to determine antimicrobial activity [18]. Chloramphenicol antibiotic (32 µg/mL) was used as the positive control. Bacterial suspension in distilled water was also used as the negative control.

#### 2.6.3. Minimum Inhibitory Concentration (MIC)

The method of Gabriele et al. [19] was used to assess the sample’s MIC against chosen microorganisms, with some modifications. To make a 5 mg/mL solution, samples were diluted in sterile water. Then, from this stock solution, dilutions of 5, 4, 3, 2, 1, 0.5, and 0.25 mg/mL were prepared using sterile distilled water. The pathogenic bacteria were cultivated in MHB for 16 h at 37 °C. The cultures were then diluted to match the turbidity of the McFarland 0.5 standard. Further dilutions in sterile MHB were performed to obtain a workable suspension containing approximately 1 × 10^−6^ CFU/mL. In a sterile 96-well plate containing 100 mL of MHB, an aliquot of 50 mL of bacterial suspensions was added. Dilutions of extracts in 100 mL were added. On each microplate well, a positive control (containing solely bacterial inoculum) was poured. The plates were incubated for 24 h at 37 °C. A microplate reader was used to detect the optical density at 600 nm. The MIC value was considered as the lowest concentration of the sample extract inhibiting the growth of the test microorganisms (no visible growth).

### 2.7. Thermogravimetric Analysis 

A thermogravimetric analyzer fitted with a temperature-controlled furnace was used to perform thermogravimetric weight loss curves (TGA, %) and derivative curves (DTG, %/C) at Chulalongkorn University’s Scientific and Technological Research Equipment Centre (STREC). In an alumina crucible, about 20 mg of the sample was inserted and heated. Temperatures ranging from 20 to 800 °C were used in the experiment, with a heating rate of 10 °C min^−1^ and a flow rate of 30 mL min^−1^ in a nitrogen atmosphere.

### 2.8. Statistical Analysis

The Box–Behnken design was used to estimate the ideal conditions from the UAE experiment using the Design Expert 11 program (Stat-Ease, Inc., Minneapolis, MN, USA). Design Expert 11 software was also used to generate three-dimensional (3D) graphs of the models. All physiochemical studies were done in triplicate, and the data were analyzed using SPSS version 20.0 statistical software and one-way analysis of variance (one-way ANOVA). Duncan’s multiple range test was used to distinguish the significant differences (*p ≤* 0.05) among the microcapsules. 

## 3. Results and Discussion

### 3.1. Optimization of Ultrasound-Assisted Extraction (UAE) Using RSM 

Table 3 shows the effect of extraction conditions (concentration, duration, and temperature) on the two distinct antioxidant activity assays (DPPH and FRAP), as well as the TFC and TPC. Table 3 shows the ANOVA results for the significance of linear, quadratic, and interaction terms of the three independent parameters (X_1_, X_2_, and X_3_) on the response values (Y). The model’s fit (Table 3) indicates whether the computed response surface accurately represents the surface’s genuine form. In three models, the lack of fit is non-significant. The R^2^ values, on the other hand, are high (0.84–0.91), indicating that the models are well-suited to the response. Table 4 also shows regression coefficient of the predicted second-order polynomial models (BBD) for bioactive properties.

Figure 1, Figure 2, Figure 3 and Figure 4 show 3D response surface plots utilized in this study. In these Figures, two independent variables were fixed and the remaining one was changed for each response to create the response surface plots. The increase in temperature boosted TPC production, which was in line with a prior study that demonstrated that heating the extracting solvent with ethanol increased TPC output [4]. According to the surface plot, an increment in the ethanol concentration (from 60% to 100% *v*/*v*) displayed lower TPC values. The ratio of ethanol to water was found to be a major determinant in extraction of phenolic compound, due to an increment in the portion of ethanol content above 60% (*v*/*v*) [20]. The TPC results of our study were in line with the mulberry leaves’ phenolic content obtained via an ultrasonic technique marked the highest level at 60% *v/v* ethanolic extraction [6]. Cisowska et al. [4] also confirmed that in comparison to only water or pure ethanol, ethanol solutions with some water, especially those with a concentration of 40–80% ethanol, were more effective at extracting polyphenolic compounds. These findings could be explained by the ease with which water and low concentrations of ethanol (i.e., 60% *v*/*v*) can enter cells to dissolve phenolic compound, as opposed to high concentrations of ethanol, which were frequently found to decrease extraction rates by causing protein denaturation, which prevents the dissolution of phenolic compound [21].

Time was another factor that had a direct impact on TPC and TFC extracted from makiang seed, according to another study from our research team, which was consistent with our current findings [14]. The extraction yield of TPC could be expressed by the following polynomial equation:TPC= −266.54118 + 6.23698X_1_ + 0.397085X_2_ + 5.32192X_3_− 0.066732X_1_∗X_2_− 0.008950X_1_∗X_3_−0.027175 X_2_∗X_3_− 0.023338 X_1_^2^ + 0.066034 X_2_^2^− 0.021185 X_3_^2^.

The linear influence of ethanol concentration on total flavonoid content was revealed to be statistically significant (*p* < 0.05) (Table 5). As shown, increasing temperature and time had a positive impact on TFC output. On the other hand, the quadratic effects and the interaction of all independent components had no effect on the TFC outcome. The extraction yield of TFC could be expressed by the following polynomial equation:TFC = −457.04271+1.82273 X_1_ +11.70707 X_2_ + 7.27436 X_3_−0.012395 X_1_∗X_2_−0.010342 X_1_∗X_3_−0.066213 X_2_∗X_3_−0.008094 X_1_^2^−0.072700 X_2_^2^−0.027467X_3_^2^.

The antioxidant activity by DPPH assay was affected by time in both linear and quadratic models (*p* ≤ 0.05). At 45 min, 55 °C, and 60% ethanol concentration, as well as at 30 min, 60 °C, and 60% ethanol concentration, respectively, the greatest and lowest DPPH values were reported (Table 2). According to Sungthong and Phadungkit [22], the highest DPPH value was obtained by extracting bioactive components from mulberry leaves at a concentration of 60% (*v*/*v*) for 25.5 min utilizing ultrasound waves, which was consistent with our findings. The response surface plots for DPPH showed that longer extraction times had a positive impact on the indicated value. The polynomial equation for DPPH assay can be described as follows:DPPH= −622.74519 − 22.58596 X_1_ + 37.65160 X_2_ + 41.93389 X_3_ +0.141317 X_1_∗X_2_ +0.105130 X_1_∗X_2_−0.310681 X_2_∗X_3_ +0.063975 X_1_^2^−0.328656 X_2_^2^−0.345514 X_3_^2^

The maximum FRAP value was found at 60 min, 55 °C, and 60% ethanol concentration and the lowest FRAP value were discovered at 30 min, 45 °C, and 100% ethanol concentration. Increasing the ethanol content from 60% to 80% resulted in lower FRAP values on the surface plot. In addition, the increase in both time (60 min) and temperature (55 °C) resulted in higher FRAP values. Consistently, it was discovered that as the temperature rose from 20 °C to 50 °C, the extraction rate increased, then fell as the temperature rose even higher [21]. High temperature speeds up mass transfer, improves a substance’s solubility, and lowers surface tension and viscosity. However, excessive heat can lead to the degradation of phenolic substances, which lowers antioxidant activity. The extracts produced at 55 °C in the current investigation had the highest levels of total phenols and antioxidant activity. These findings were consistent with Akowuah et al. [23]’s finding that the level of total phenolics did not alter significantly between extraction temperatures of 40 °C and 50 °C. However, from extraction temperatures of 60 °C and above, a drop in total phenolic content occurred. Due to chemical and enzymatic degradation brought on by the high temperature used during extraction, polyphenols become less stable and lose their antioxidative properties. Additionally, the quantity of antioxidants in an extract may be reduced by the presence of polyphenol oxidases. The polyphenol oxidase may have been activated quickly enough to start the destruction of the markers at an extraction temperature higher than 60 °C. Although polyphenol metabolites have significant antioxidant effects, they can breakdown into another substance that has no antioxidant activity at high temperatures [23]. The FRAP assay model for antioxidant activity is represented by the equation below:FRAP = −866.90861+16.73041 X_1_ +26.47131X_2_ +24.16231X_3_−0.068382 X_1_∗X_2_ +0.036617 X_1_∗X_3_−0.218217 X_2_∗X_3_−0.145391X_1_^2^−0.053910 X_2_^2^−0.121271 X_3_^2^.

### 3.2. Antimicrobial Activity by Disk Diffusion Assay and Minimum Inhibitory Concentration (MIC) 

The CSP microcapsules preventing the growth of bacteria were tested, including *Bacillus subtilis*, *Escherichia coli*, *Salmonella Typhimurium*, and *Staphylococcus aureus*. However, the GMM had the largest inhibitory zones against *Escherichia coli* and *Staphylococcus aureus* among the other samples (11.5 and 11.1 mm, respectively). In terms of antimicrobial activity (inhibition zone and MIC), there was an insignificant difference (*p* > 0.05) between GMM and CSP extract. Furthermore, in terms of inhibitory zone diameter, no significant difference (*p* > 0.05) was observed between GMM and Chloramphenicol (positive control). According to a recent study, encapsulation of cell-free *Lactobacilli* extracts and ethanolic *propolis* extracts with maltodextrin as the wall material resulted in higher inhibition zones against foodborne pathogens [24].

*Escherichia coli* and *Staphylococcus aureus* were the most sensitive microorganisms to CSP microcapsules, as determined by the lowest MIC values (Table 6). Gram-negative bacteria function as barriers to foreign materials because they have an outer membrane and periplasmic space that Gram-positive bacteria do not [25]. GMM had the lowest MIC value (1.50), followed by GAM (2.00) and MDM (2.25). Our results supported the findings of our previous publication that makiang seed extracts were effective against *Staphylococcus aureus* [14]. According to Karaaslan et al. [26], the pepper seed oil’s antibacterial potential was enhanced by GMM microencapsulation by spray-drying. The results of our investigation and other studies regarding antimicrobial activity could be attributed to the presence of phenolic compound present in the tested samples that have exceptional antibacterial activity [25]. Başyiği et al. [25] emphasized that to prevent the spread of pathogens in foods or in the human diet, microencapsulation, a technology that helps in dispersing and preserving functional food-value compounds within the food matrix, could be utilized. Donsì et al. [27] provided evidence to support this claim by showing that nanoencapsulation of bioactives (such as essential oils) could increase their antibacterial activity in foods. However, according to them, the kind of microorganisms and the formulation both affect how much antimicrobial activity increases.

### 3.3. Effects of Microencapsulation on Functional Properties of CSP 

The TPC, TFC, DPPH and FRAP among the CSP microcapsules (GAM, MDM and GMM) were found to be in the range of 140.20–169.09 (mg GAE/100g dry wt.), 88.21–114.69 (mg QE/100g dry wt.), 705.81–1063.00 (mM trolox/100 g dry wt.) and 1180.84–1202.95 (mM trolox/100 g dry wt.), respectively (Figure 5). Using CSP microcapsules with the mixtures of wall materials (i.e., GMM) produced the higher DPPH values (*p* ≤ 0.05) comparing with each microcapsule alone (i.e., GAM and MDM), indicating that they had synergistic effects and provided better protection to these compounds during the spray-drying process [9,28]. One of the most popular and frequently utilized natural materials for microencapsulation is MD. It is a polysaccharide, and because of its strong water solubility, it is very convenient to employ in the microencapsulation process by spray-drying [29]. Using GA as a wall material and as an encapsulating agent in the manufacture of honey powder, amla extract and basil leaves increased the phenolic content and antioxidant activity, according to Rajabi et al. [11]. Nambiar et al. [30], who obtained comparable findings in the microencapsulation of tender coconut water by spray-drying, supported our higher results in terms of antioxidant activities (DPPH assay) in microcapsule powders. They discovered that increasing the wall material concentration enhanced the bioactivity of the powders, and that spray-drying was a promising methodology for encapsulating phenolic chemicals for long-term protection and controlled release under particular circumstances. According to Yekdane and Hossein Goli [5], encapsulation with pomegranate juice and GA and xanthan gum as the wall materials was successful in enhancing the oxidative stability of pomegranate seed oil during storage, which ensures future studies on the stability storage of CSP microcapsules.

### 3.4. Quality Characteristics of Microcapsules

The microcapsules’ physical features did not differ significantly (*p* > 0.05). Encapsulation efficiency, moisture content, water activity, and solubility of the microcapsules fell into the ranges of 59.92–67.05%, 2.56–2.72%, 0.10–0.15%, and 75.50–75.99%, respectively. One of the primary factors affecting the quality and shelf life of the microcapsules are moisture content and water activity. Due to the reduced water activity found in this study, powders may be more resistant to degrading reactions, including enzymatic and non-enzymatic browning during storage. On the other hand, the high-water activity is linked to an increase in the number of bacteria that contaminate food, specifically the growth of microbes that produce toxins that may be detrimental to people [12]. The moisture content of the CSP microcapsules was less than 5% during our examination, which was consistent with our earlier findings [6]. In line with this work, we also validated in a recent review publication that bioactive encapsulation by spray-drying showed advantageous physicochemical characteristics throughout the course of a decade of research, including low moisture, water activity, small particle size, and good solubility [8]. However, a decade of research revealed that the wall/core ratio, carrier composition, and type were crucial in the encapsulation and maintenance of physicochemical, structural, and functional characteristics of microcapsules. The initial composition of the raw material, encapsulating agents, and the spray-dry condition (e.g., compressed air flow, and feed flow) were also only a few of the variables that might impact the solubility of microcapsules. In a study on the spray-drying encapsulation of yacon juice, the high solubility of the microcapsules (>75%) in our analysis was validated. Color values were unaffected by encapsulation types (*p* > 0.05). The L*, a*, and b* values varied from 60.39–66.46, 1.00–1.14, and 9.91–14.11, respectively.

### 3.5. Microcapsule’s Surface Structure Characterization and Particle Size Distribution

An important aspect that impacts the appearance, flowability, and dispensability of powders is particle size. GAM, MDM, and GMM had particle sizes ranging from 8 to 80 µm, 6 to 110 µm, and 11 to 80 µm, respectively (Figure 6). GAM, MDM, and GMM had D (4,3) (volume mean diameter) and D (3,2) (surface weighed mean) of 31.00 and 15.50 µm, 31.90 and 14.60 µm, and 34.30 and 18.50 µm, respectively. The relationships between the input temperature, increased extract feed rate, type of encapsulating agent, and feed composition may have an impact on the particle size [31]. The particle size distribution of the microcapsules in the current investigation did not show any significant changes (*p* > 0.05); nevertheless, Colín-Cruz et al. [32] indicated that the combination of high- and low-viscosity materials resulted in particles with sizes between the values of separate components which was the case for GMM in this study.

Preferable microcapsules have a smooth surface, a spherical shape, and no dents or pleats. The spherical shape of the microcapsules (GAM, MDM, and GMM) has been identified in Figure 7. Consistently, the surface of pepper seed oil microcapsules with gum arabic/maltodextrin as the wall material displayed the same characteristic [26]. Spray-drying caused the dents on the microcapsules by creating a pressure of steam on the interior structure, causing rapid shrinking of the surface owing to moisture loss [13]. The smooth surface and shape of the microcapsules can be attributed to the 2:1 wall-to-core ratio used in this study, which resulted in a thicker encapsulation layer, and a smoother surface and shape. When different microcapsules (GAM, MDM, and GMM) were compared, it was observed that they were essentially identical in shape, but for a few slight dents and pleats. Similarly, encapsulating blackberry juice powder with resistant maltodextrin and gum arabic created spherical microcapsules with a smooth surface [33].

### 3.6. Analyses of Thermogravimetry 

Thermogravimetric (TGA) and differential thermogravimetric (DTG) curves provide sufficient information about a material’s thermal behavior and current data on its thermal stability and composition, as well as assisting in estimating the examined material’s maximum weight loss or gain as a function of temperature [26]. TGA and DTG were used to measure the weight change of the microparticles and wall materials as a function of temperature in this study. The TGA and DTG are shown in Figure 8 in relation to the weight loss intensity during heat depolymerization. Particularly for GAM and GMM, the microcapsules responded almost identically. The mass loss in GAM, MDM, and GMM during the first heating phase, up to 140 °C, was 4.97, 3.61, and 5.83%, respectively, indicating that unencapsulated volatile chemicals and unbound water were the sources of mass loss in the samples. After this first phase, the MDM showed a 21.11% mass loss at 240 °C, possibly reflecting the loss of unprotected volatile chemicals. GAM and GMM, on the other hand, lost 59.13 and 63.71% of their mass, respectively, when heated to 400 °C. Phenolic volatilization was presumably hindered by the interaction of phenolic chemicals with the polymeric covering material. The MDM lost 63.64% of its mass when the temperature was elevated to 800 °C, although the GAM and GMM only lost 28.13 and 21.26% of their bulk, respectively. The disintegration of the fundamental constituents of the wall materials may have prompted the deterioration of the phenolic compound dispersed throughout the matrix of the powders. These microcapsules have a lot of potential for food processing applications, according to TGA and DTG tests. They showed significant thermal stability when exposed to high temperatures frequently used in the food sector without any protection.

## 4. Conclusions

This work successfully optimized the extraction conditions for bioactive chemicals (TPC and TFC) and antioxidant properties (DPPH and FRAP assays) from CSP using the ultrasound-assisted extraction (UAE) technique optimized by RSM. It was determined that 55 °C, 45 min, and 60% ethanol concentration were the optimum extraction conditions. The physicochemical, morphological, thermal, and bioactive properties of CSP microencapsulated by spray-drying were measured. The microcapsules’ physical features were insignificant and the microcapsule’s surface structure was identical in shape, with only a few slight dents and pleats. The results also showed that CSP performed better when it was encapsulated in a GA and MD mixture (GMM) in terms of antibacterial (higher inhibition zone and lower MIC against *Bacillus subtilis*, *Escherichia coli*, *Salmonella Typhimurium*, and *Staphylococcus aureus*) and antioxidant activities (DPPH: mM trolox/100 g dry wt.). In conclusion, the encapsulation by spray-drying could be considered as a viable technology to produce functional food ingredients; however, future studies should concentrate on using CSP microcapsules, in particular GMM, as an enrichment component in food and food packaging formulations.

## Figures and Tables

**Figure 1 foods-12-00412-f001:**
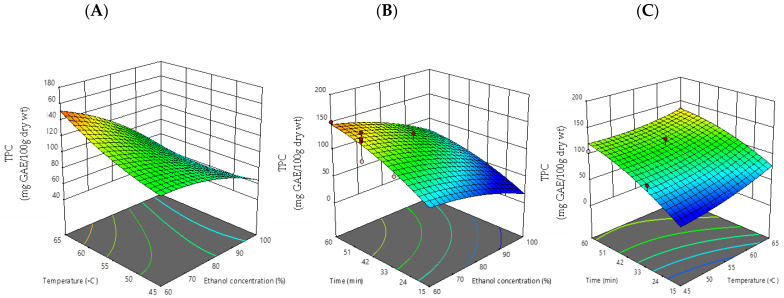
Response surface plots (3D) of total phenolic compound (TPC) as a function of significant interaction between factors; (**A**) temperature and ethanol concentration; (**B**) time and ethanol concentration; (**C**) time and temperature of CSP extract.

**Figure 2 foods-12-00412-f002:**
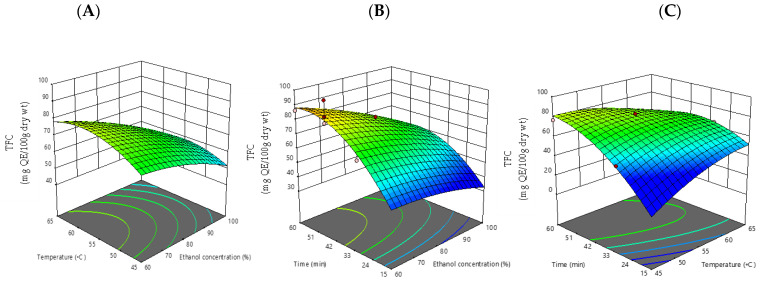
Response surface plots (3D) of total flavonoid content (TFC) as a function of significant interaction between factors;(**A**) temperature and ethanol concentration; (**B**) time and ethanol concentration; (**C**) time and temperature of CSP extract.

**Figure 3 foods-12-00412-f003:**
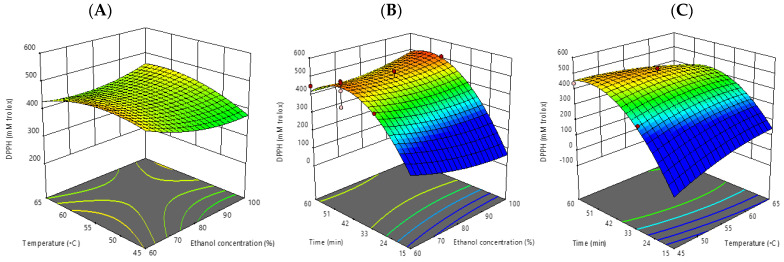
Response surface plots (3D) of antioxidant activity (DPPH assay) as a function of significant interaction between factors;(**A**) temperature and ethanol concentration; (**B**) time and ethanol concentration; (**C**) time and temperature of CSP extract.

**Figure 4 foods-12-00412-f004:**
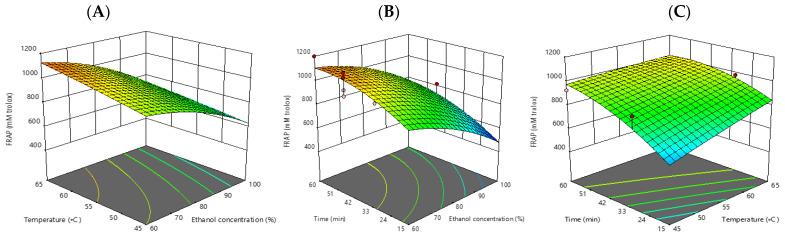
Response surface plots (3D) of antioxidant activity (FRAP assay) as a function of significant interaction between factors; (**A**) temperature and ethanol concentration; (**B**) time and ethanol concentration; (**C**) time and temperature of CSP extract.

**Figure 5 foods-12-00412-f005:**
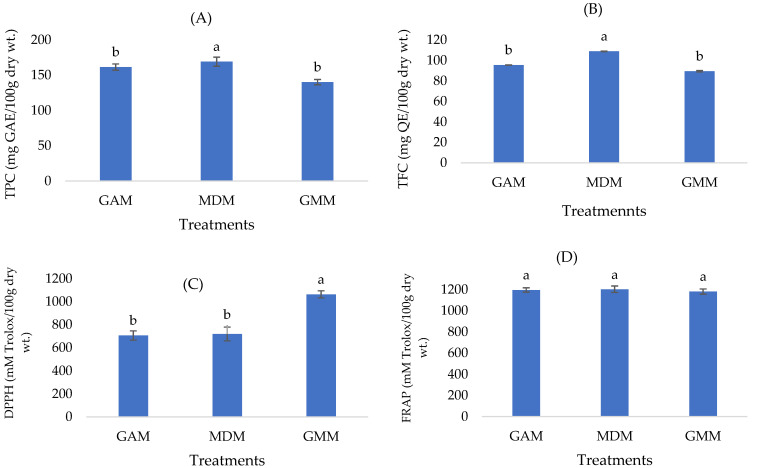
TPC: Total phenol compound (**A**), TFC: total flavonoid content, TFC (**B**), DPPH: 2,2-Diphenyl-1-picrylhydrazyl (**C**) and FRAP: Ferric Reducing Antioxidant Power (**D**) of CSP microcapsules. GAM: gum arabic microcapsule (CSP extract coated with GA), MDM: maltodextrin microcapsule (CSP extract coated with MD), GMM: gum arabic + maltodextrin microcapsule (CSP extract coated with the mixture of GA + MD). The letters “a, b” show the significant differences (*p* ≤ 0.05).

**Figure 6 foods-12-00412-f006:**
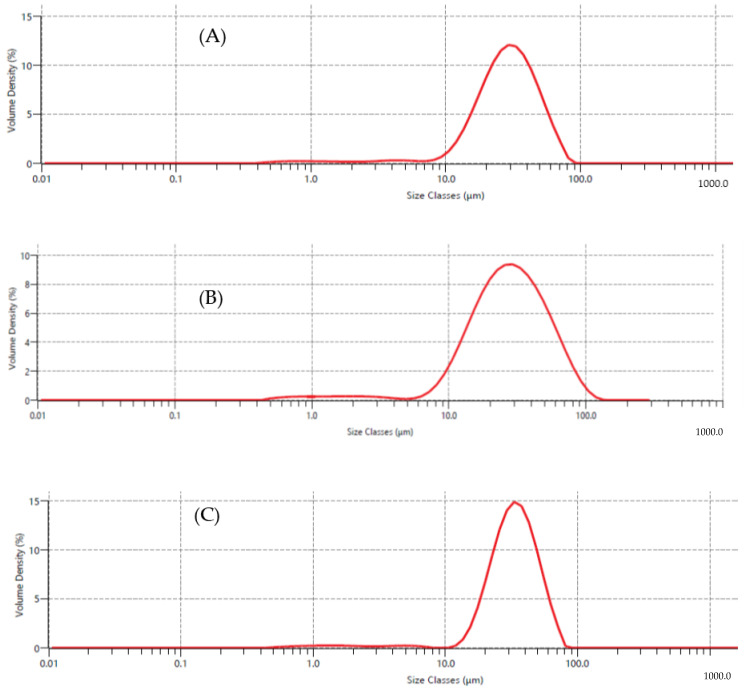
Particle size distribution of microcapsules. (**A**) GAM, gum arabic microcapsule (CSP extract coated with GA), (**B**) MDM, maltodextrin microcapsule (CSP extract coated with MD), and (**C**) GMM, gum arabic + maltodextrin microcapsule (CSP extract coated with the mixture of GA + MD).

**Figure 7 foods-12-00412-f007:**
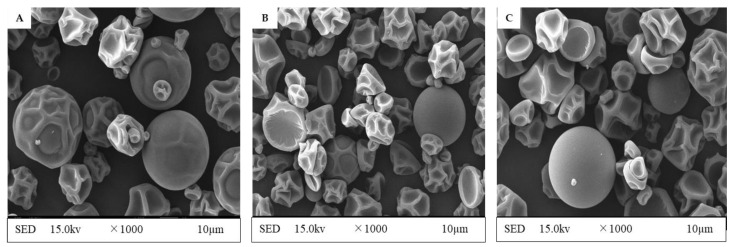
Scanning electron microscopy at ×1000 magnification of CSP extract microcapsules. (**A**) GAM, gum arabic microcapsule (CSP extract coated with GA), (**B**) MDM, maltodextrin microcapsule (CSP extract coated with MD), (**C**) GMM, gum arabic + maltodextrin microcapsule (CSP extract coated with the mixture of GA + MD).

**Figure 8 foods-12-00412-f008:**
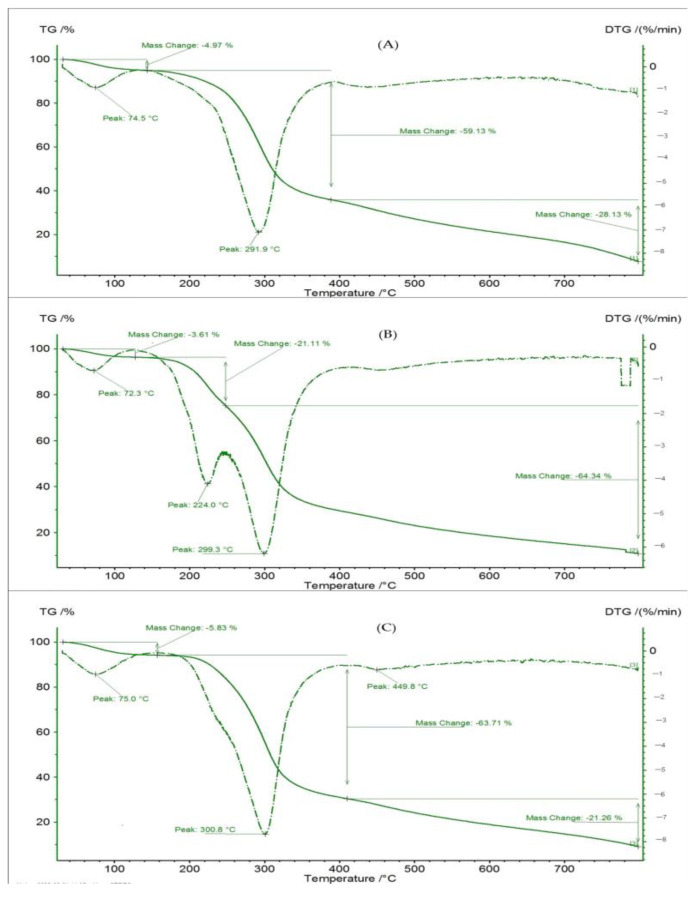
TG (%, continuous line) and DTG (%/min, discontinuous line) curves of the microcapsules. (**A**) GAM, gum arabic microcapsule (CSP extract coated with GA), (**B**) MDM, maltodextrin microcapsule (CSP extract coated with MD), (**C**) GMM, gum arabic + maltodextrin microcapsule (CSP extract coated with the mixture of GA+MD).

**Table 1 foods-12-00412-t001:** The values for the Box–Behnken design (BBD).

Independent Variables	Code Symbols	Level		
		−1	0	1
Concentration (%)	X_1_	60	80	100
Temperature (°C)	X_2_	45	55	65
Time (min)	X_3_	30	45	60

**Table 2 foods-12-00412-t002:** The functional properties of cocoa shell powder (CSP) extract under different extraction conditions.

	Independent Variables			Responses			
Run	Concentration (%)	Temperature (°C)	Time (min)	TPC (mg GAE/100g dry wt)	TFC (mg QE/100g dry wt)	DPPH (mM Trolox/100g dry wt)	FRAP (mM Trolox/100g dry wt)
1	60	55	45	140.50	84.37	522.38	1134.53
2	60	65	45	160.12	74.95	450.00	1197.68
3	60	55	30	97.22	67.93	424.25	980.21
4	60	55	45	100.25	99.26	480.00	1098.20
5	60	55	45	150.70	88.58	530.00	1000.58
6	80	45	60	105.98	76.85	438.00	926.11
7	100	45	30	45.69	41.07	246.13	479.78
8	80	65	30	102.26	65.48	353.63	978.74
9	60	55	60	150.38	86.20	448.00	1197.68
10	80	55	45	119.92	78.64	502.00	951.89
11	80	45	30	87.30	52.93	319.25	885.00
12	60	55	45	135.90	84.23	393.63	951.37
13	100	65	30	63.71	53.74	398.63	679.00
14	100	45	60	104.40	69.64	495.50	785.00
15	80	65	60	132.56	57.43	320.50	959.79
16	60	45	45	110.22	77.00	512.00	1024.00
17	100	55	45	53.92	52.34	509.34	820.00

Total phenolic compound (TPC), total flavonoid content (TFC), antioxidant activity by 2,2-Diphenyl-1-picrylhydrazyl(DPPH) and ferric reducing antioxidant power(FRAP) assays.

**Table 3 foods-12-00412-t003:** Analysis of variance (ANOVA) for determination of optimization model fit.

		TPC (mg GAE/100g dry wt)				TFC (mg QE/100g dry wt)				DPPH (mM Trolox/100 g dry wt)				FRAP (mM Trolox/100g dry wt)		
Source	Sum of Squares	df	Mean Squares	*p*-Value	Sum of Squares	df	Mean Squares	*p*-Value	Sum of Squares	df	Mean Squares	*p*-Value	Sum of Squares	df	Mean Squares	*p*-Value
**Model**	3538.27	9	1770.20	0.0246	93,693.10	9	393.14	0.0055	15,931.78	9	10,410.34	0.0330	4.625	9	51,392.22	0.0202
**X_1_**	725.39	1	5580.32	0.0058	996.45	1	725.39	0.0059	5580.32	1	996.45	0.5398	1.965	1	1.965	0.0029
**X_2_**	39.98	1	1224.16	0.1096	960.29	1	39.98	0.3909	1224.16	1	960.29	0.5471	34,191.45	1	34,191.45	0.1047
**X_3_**	261.12	1	1089.46	0.1276	25,346.69	1	261.12	0.0521	1089.46	1	25,346.69	0.0140	16,889.82	1	16,889.82	0.2316
**X_1_X_2_**	22.90	1	663.85	0.2193	2977.05	1	22.90	0.5112	663.85	1	2977.05	0.3021	697.09	1	697.09	0.7978
**X_1_X_3_**	35.87	1	26.87	0.7939	3707.12	1	35.87	0.4151	26.87	1	3707.12	0.2539	449.72	1	449.72	0.8369
**X_2_X_3_**	509.21	1	85.77	0.6425	11,210.95	1	509.21	0.0138	85.77	1	11,210.95	0.0674	5530.83	1	5530.83	0.4780
**X_1_^2^**	27.07	1	225.01	0.4580	1690.84	1	27.07	0.4763	225.01	1	1690.84	0.4289	8732.85	1	8732.85	0.3777
**X_2_^2^**	139.45	1	115.05	0.5919	2849.84	1	139.45	0.1314	115.05	1	2849.84	0.3118	76.68	1	76.68	0.9322
**X_3_^2^**	100.77	1	59.94	0.6973	15,945.32	1	100.77	0.1899	59.94	1	15,945.32	0.0366	1964.34	1	1964.34	0.6686
**Residual**	334.69	7	364.79		16,793.65	7	47.81		2553.52	7	2399.09		68,924.78	7	9846.40	
**Lack of Fit**	185.11	4	277.10	0.7029	5046.83	4	46.28	0.5467	1108.41	4	1261.71	0.8487	47,344.69	4	11,836.17	0.3557
**Pure Error**	149.59	3	481.70		11,746.81	3	49.86		1445.11	3	3915.60		21,580.10	3	7193.37	
**Cor Total**	3872.96	16			1.105	16			18,485.29	16			5.315	16		
**C.V.%**				17.46				9.70				11.34				10.51
**R^2^**				0.86				0.91				0.84				0.87
**Adj-R^2^**				0.68				0.80				0.65				0.70

Ethanol concentration (X_1_), Temperature (X_2_), Time (X_3_).

**Table 4 foods-12-00412-t004:** Regression coefficient of the predicted second-order polynomial models (BBD) for bioactive properties.

	Bioactive Properties			
Factor	TPC	TFC	DPPH	FRAP
Intercept	97.91 *	72.98 *	439.52 *	928.31 *
**Linear**				
X_1_	−30.06 *	−10.84 *	−12.70	−178.40 *
X_2_	13.03	2.36	11.54	68.88
X_3_	34.25	16.77	165.23 *	134.87
**Cross product**				
X_1_X_2_	−13.35	−2.48	28.26	−13.68
X_1_X_3_	−4.03	−4.65	47.31	16.48
X_2_X_3_	−6.11	−14.90 *	−69.90	−49.10
**Quadratic**				
X_1_^2^	−9.34	−3.24	25.59	−58.16
X_2_^2^	6.60	−7.27	−32.87	−5.39
X_3_^2^	−10.72	−13.91	−174.92 *	−61.39

* Level of significance *p* ≤ 0.05. Ethanol concentration (X_1_), Temperature (X_2_), Time (X_3_).

**Table 5 foods-12-00412-t005:** Effects of encapsulation by spray-drying on quality properties of microcapsules.

	Treatments				
Parameters	GAM	MDM	GMM	SEM	*p*-Value
Encapsulation efficiency (%) ^ns^	67.05 ± 0.37	62.95 ± 1.98	59.92 ± 0.44	0.973	1.020
Moisture content (%) ^ns^	2.72 ± 0.01	2.56 ± 0.07	2.69 ± 0.07	0.012	3.090
Water activity ^ns^	0.13 ± 0.00	0.15 ± 0.00	0.10 ± 0.00	0.003	3.900
Solubility (%) ^ns^	75.99 ± 0.14	75.94 ± 0.06	75.50 ± 0.55	0.073	0.359
**Color values ^ns^**					
L*	66.46 ± 0.00	60.39 ± 0.00	63.79 ± 0.07	0.827	1.330
a*	1.00 ± 0.00	1.14 ± 0.00	1.09 ± 0.00	0.018	1.930
b*	14.11 ± 0.00	9.91 ± 0.00	11.92 ± 0.00	0.573	3.960
**Particle size (μm) ^ns^**					
D (4,3)	31.00 ± 0.06	31.90 ± 0.01	34.30 ± 0.03	0.464	1.180
D (3,2)	15.50 ± 0.02	14.60 ± 0.01	18.50 ± 0.04	0.555	1.730

The superscript “^ns^” shows non-significance. Three replications were used for each microcapsule per each analysis. GAM: gum arabic microcapsule (CSP extract coated with GA), MDM: maltodextrin microcapsule (CSP extract coated with MD), GMM: gum arabic + maltodextrin microcapsule (CSP extract coated with the mixture of GA + MD). SEM: standard error of mean. D (4,3) is the mean diameter over volume and D (3,2) is volume/surface mean.

**Table 6 foods-12-00412-t006:** Antibacterial activity of CSP extract microcapsules from the optimized ultrasound-assisted extraction.

		Microorganism		
Sample/Parameter	*Bacillus subtilis*	*Escherichia coli*	*Salmonella Typhimurium*	*Staphylococcus aureus*
**Inhibition zone (mm)**				
CSP extract	9.4 ± 0.20 ^a^	11.3 ± 0.50 ^a^	8.9 ± 0.75 ^b^	10.9 ± 0.20 ^a^
GAM (5 mg/mL)	7.1 ± 0.35 ^c^	10.05 ± 0.10 ^b^	7.2 ± 0.20 ^c^	9.8 ± 0.20 ^b^
MDM (5 mg/mL)	8.4 ± 0.15 ^b^	8.9 ± 1.20 ^c^	8.1 ± 0.34 ^c^	8.5 ± 0.25 ^c^
GMM (5 mg/mL)	9.6 ± 0.15 ^a^	11.5 ± 0.34 ^a^	9.0 ± 0.50 ^b^	11.1 ± 0.45 ^a^
Chloramphenicol (30 µg/mL)	9.5 ± 0.20 ^a^	11.2 ± 0.30 ^a^	11.0 ± 0.50 ^a^	9.1 ± 0.16 ^b^
**MIC (mg/mL)**				
CSP extract	1.80 ^b^	1.65 ^b^	1.90 ^b^	1.50 ^b^
GAM	2.85 ^a^	2.00 ^a^	2.85 ^a^	2.00 ^a^
MDM	2.25 ^a^	2.50 ^a^	2.25 ^a^	2.50 ^a^
GMM	1.85 ^b^	1.50 ^b^	1.85 ^b^	1.50 ^b^

Three replications were used for each microcapsule per microorganism. CSP: cocoa shell powder, GAM: gum arabic microcapsule (CSP extract coated with GA), MDM: maltodextrin microcapsule (CSP extract coated with MD), GMM: gum arabic + maltodextrin microcapsule (CSP extract coated with the mixture of GA+MD). Chloramphenicol solution was used as positive control. Different letters (a, b and c) in each column show significant differences (*p* ≤ 0.05).

## Data Availability

Not applicable.

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
