# Peer review of "Ultrasound-Assisted Extraction of Bioactive Compounds from Cocoa Shell and Their Encapsulation in Gum Arabic and Maltodextrin: A Technology to Produce Functional Food Ingredients"

_foods, 2023, doi:10.3390/foods12020412_

Round 1

Reviewer 1 Report

The Authors are advised to make the following corrections:

1) Rephrase the line no 40-43. No need to start the line from Because ot its............. The sentence can be framed as - On the account of helath benefits etc.................

2) Rephrase the line no's 48 - 49.

3) Rephrase the line no's 50-51 as UAE has become a popular technique for extraction of essential componenets from naturalmatrices.....

4)Rephrase the line no's  52-54: Sevral researchers have adopted RSM for the optimization of UAE.

5)Line no 57-59 - not clear.

6) Rephrase teh lien no's 65 - 68.

7)Line no 83 -word response & RSM can't be used it has to be RSM only.

8) line no 84 - the word is making, please correct.

9) line no 85 - 86 : wrong way of starting the sentenses.

10)line no 91-92 again wrong

11) line no107; please use abbreviationas UAE.

12) line no116, 133 - we can't write 'described by our's'. 

13) line no 131 - please use word kept not out

14) Line no 142:  no specifications missing

15) Line no 146: it should be honogenized not mixed

16)  Line no 164: please remove the extra dot

A lot many such mistakes are there in the manuscript. Authors are advised to please re-check the complete manucript carefully.

17) In the text sometimes, 3-D and many times 3-Dimensional words are mentioned. Plesae use one common way. Follow the same formats for using symbols like % in the text. Degree symbols are also not well checked. 

18) Line No 344 & 412: GMM has alredy been used. now why authors are using the words gum Arabic... the full form. Please use same pattern.

19) Line no 1407-408: Re-check.  

Conclusion part please recheck and re-write.

Author Response

Dear Editor-in-chief,                                                                           

We would like to thank you and the reviewers for the valuable and useful comments given with regards to our manuscript entitled "Ultrasound-assisted extraction of bioactive compounds from cocoa shell and their encapsulation in gum Arabic and maltodextrin: A technology to produce functional food ingredients."

We have made the following corrections/modifications and additions to our manuscript. We also addressed the specific comments raised by the reviewers and the detailed responses are listed out in the following table.

Following revision of the manuscript taking into account all the comments and suggestions of the reviewers, it is hoped that the quality of the manuscript has improved and can now be considered for publication. Thank you.

Sincerely,

Authors

NO.

Comments from the  1st reviewer

Corrections Made

-

Extensive editing of English language and style required.

The other 2 reviewers have suggested only “spell check”. Besides, this manuscript has been revised

by a native English writer.

1

Rephrase the line no 40-43. No need to start the line from Because ot its............. The sentence can be framed as - On the account of helath benefits etc.................

Change has been made as requested.

2

Rephrase the line no's 48 - 49.

Done as requested.

3

Rephrase the line no's 50-51 as UAE has become a popular technique for extraction of essential componenets from naturalmatrices.....

Done as requested.

4

Rephrase the line no's  52-54: Sevral researchers have adopted RSM for the optimization of UAE.

Done as requested.

5

Line no 57-59 - not clear.

The lines “57-59” have been deleted due to redundancy.

6

Rephrase the line no's 65 - 68.

Done as requested.

7

Line no 83 -word response & RSM can't be used it has to be RSM only.

“response” has been deleted.

8

line no 84 - the word is making, please correct.

makiang seed” has been inserted.

9

line no 85 - 86 : wrong way of starting the sentenses.

Line 85-86 has been paraphrased.

10

line no 91-92 again wrong

Line 91-92 has been paraphrased.

11

line no107; please use abbreviationas UAE.

Done as requested.

12

line no116, 133 - we can't write 'described by our's'.

The line “116-133” have been rechecked.

13

line no 131 - please use word kept not out

Done as requested.

14

Line no 142:  no specifications missing

Done as requested.

15

Line no 146: it should be honogenized not mixed

“Homogenized” has been inserted.

16

Line no 164: please remove the extra dot

Done as requested.

17

In the text sometimes, 3-D and many times 3-Dimensional words are mentioned. Plesae use one common way. Follow the same formats for using symbols like % in the text. Degree symbols are also not well checked.

Done as requested.

18

Line No 344 & 412: GMM has alredy been used. now why authors are using the words gum Arabic... the full form. Please use same pattern.

Done as requested.

19

Line no 407-408: Re-check. 

Line 407-408 has been rechecked.

20

Conclusion part please recheck and re-write.

Done as requested.

Reviewer 2 Report

The manuscript discusses the Ultrasound-assisted extraction of bioactive compounds from a cocoa shell.

1.       Figures 4-8 are missing

2.       It is very difficult to follow Table 1-6 because it contains many columns and numbers that mostly relate to statistical calculation. I recommend replacing it with a more informative table or transforming it into figures. Tables 1-6 could be as supporting data.

3.       As figures 4-8 are missing, I can not evaluate the result.

4.       Some results were evaluated only based on the previous result without any rigorous explanation of the phenomenon (such as section 3.3). More discussion on it will be suggested in the revised manuscript.

Author Response

Dear Editor-in-chief,                                                                           

We would like to thank you and the reviewers for the valuable and useful comments given with regards to our manuscript entitled "Ultrasound-assisted extraction of bioactive compounds from cocoa shell and their encapsulation in gum Arabic and maltodextrin: A technology to produce functional food ingredients."

We have made the following corrections/modifications and additions to our manuscript. We also addressed the specific comments raised by the reviewers and the detailed responses are listed out in the following table.

Following revision of the manuscript taking into account all the comments and suggestions of the reviewers, it is hoped that the quality of the manuscript has improved and can now be considered for publication. Thank you.

Sincerely,

Authors

NO.

Comments from the  2nd  reviewer

Corrections Made

1

Figures 4-8 are missing

Figures 4-8 are included now.

2

It is very difficult to follow Table 1-6 because it contains many columns and numbers that mostly relate to statistical calculation. I recommend replacing it with a more informative table or transforming it into figures. Tables 1-6 could be as supporting data.

Thank you for your commnet. However, in most cases and through the literature, the RSM results will be shown and presented in the current format.

There are many examples, but this one is one of thouse articles you can kindly refer to:

https://doi.org/10.1016/j.ultsonch.2021.105806.

3

As figures 4-8 are missing, I can not evaluate the result.

Figures 4-8 are added now.

4

Some results were evaluated only based on the previous result without any rigorous explanation of the phenomenon (such as section 3.3). More discussion on it will be suggested in the revised manuscript.

Some improvements have been made in this section.

Reviewer 3 Report

·         Reviewers comments: In introduction section, the authors must define microencapsuñation by spray-spray drying.

 ·         Reviewers comments: It is still difficult to find the novelty of the work concerning what has already been published. What is the difference between what is published with what the authors want to publish? It is not clear. The authors must describe these differences in the introduction section.

 ·         Reviewers comments: The authors must displayed the image of size and shape of the microparticles obtained by SEM. The manuscript did not show these results.

 ·         Reviewers comments: Line 386: “The microcapsules' physical features did not differ significantly (P>0.05)” .  Is p<0.05 or p>0.05????. Please revise.

 ·         Reviewers comments: The authors must display the TGA results.

Author Response

Dear Editor-in-chief,                                                                           

We would like to thank you and the reviewers for the valuable and useful comments given with regards to our manuscript entitled "Ultrasound-assisted extraction of bioactive compounds from cocoa shell and their encapsulation in gum Arabic and maltodextrin: A technology to produce functional food ingredients."

We have made the following corrections/modifications and additions to our manuscript. We also addressed the specific comments raised by the reviewers and the detailed responses are listed out in the following table.

Following revision of the manuscript taking into account all the comments and suggestions of the reviewers, it is hoped that the quality of the manuscript has improved and can now be considered for publication. Thank you.

Sincerely,

Authors

NO.

Comments from the  3rd   reviewer

Corrections Made

1

In introduction section, the authors must define microencapsuñation by spray-spray drying

The 3rd paragraph in the introduction section eplains thoroughly about microencapsulation:

Using carriers (such as maltodextrin, gum Arabic, etc.) that can prevent diges-tion-related degradation, increase bioactivity and bioavailability, and support both regu-lated release and targeted administration, microencapsulation is a promising method for increasing the distribution of bioactive ingredients in foods [8]. Microencapsulation of ac-tive substances is accomplished in food application contexts using a variety of processes, including fluidized bed coating, inclusion complexation, complex coacervation, freeze drying, spray drying and extrusion [9]. However, the most popular encapsulation tech-nique among these known technologies is spray drying since it is affordable, simple to apply, and produces high-quality particles [7]. Choosing the right wall material/ carrier is crucial for microencapsulation. Only food-grade materials that have been certified as Generally Recognized as Safe (GRAS) or permitted by regulatory authorities like the Food and Drug Administration (FDA) and the European Food Safety Authority (EFSA) may be used for wall applications. Due to their high solubility, outstanding biocompatibility and safety, maltodextrin (MD) and gum Arabic (GA) are among the most frequently utilized polymers that have been used as carriers for the spray-drying encapsulation of bioactive chemicals [10]. In the recent decade, there were numerous studies about the characteriza-tions of microcapsules obtained from encapsulation by spray drying from different bioac-tive sources [11, 12, 13]. Even in our recent research studies, we confirmed the encapsula-tion of bioactives from mulberry leaf by spray drying [6] and also used RSM to enhance the extraction of bioactive components from makiang seed for use in orange juice [14]. In addition, using spray drying to microencapsulate bioactive compounds in fruit products could open up new possibilities for the controlled delivery of beneficial compounds to the host, as well as a potentially cost-effective method to prevent and/or treat a wide range of diseases. However, until today, there is no information in the literature about the effects of the above-mentioned encapsulation in the extract from cocoa shell powder (CSP).”

2

It is still difficult to find the novelty of the work concerning what has already been published. What is the difference between what is published with what the authors want to publish? It is not clear. The authors must describe these differences in the introduction section.

Thank you for your commnet. Although there are some studies on application of RSM and encapsulation on some bioactives, however, until now there is no information in the literature about the effects of the above-mentioned extraction and encapsulation in cocoa shell powder (CSP). That’s why the lack of information on CSP was the most incentive to conduct the current study.

3

The authors must displayed the image of size and shape of the microparticles obtained by SEM. The manuscript did not show these results.

Done as requested.

However, an enough explannation has been inserted in the Figure 7.

4

Line 386: “The microcapsules' physical features did not differ significantly (P>0.05)” .  Is p<0.05 or p>0.05????. Please revise.

Done as requested.

5

The authors must display the TGA results.

The results have been displayed in Figure 8 and in the section related to TGA.

Round 2

Reviewer 2 Report

The manuscript has been well corrected/revised. The point that I highlight is only the presence of Table 4 in not mentioned in the text. Moreover, the explanation of the result is already in good form.